# Parallelizing Linear Recurrent Neural Nets Over Sequence Length

**Eric Martin**
eric@ericmart.in

**Chris Cundy**
Department of Computer Science
University of California, Berkeley
Berkeley, CA 94720, USA*
c.cundy@berkeley.edu

## Abstract

Recurrent neural networks (RNNs) are widely used to model sequential data but their non-linear dependencies between sequence elements prevent parallelizing training over sequence length. We show the training of RNNs with only linear sequential dependencies can be parallelized over the sequence length using the parallel scan algorithm, leading to rapid training on long sequences even with small minibatch size. We develop a parallel linear recurrence CUDA kernel and show that it can be applied to immediately speed up training and inference of several state of the art RNN architectures by up to 9x. We abstract recent work on linear RNNs into a new framework of linear surrogate RNNs and develop a linear surrogate model for the long short-term memory unit, the GILR-LSTM, that utilizes parallel linear recurrence. We extend sequence learning to new extremely long sequence regimes that were previously out of reach by successfully training a GILR-LSTM on a synthetic sequence classification task with a one million timestep dependency.

## 1 Introduction

Recurrent neural networks (RNNs) are widely used for sequence modelling tasks in domains such as natural language processing (Sutskever et al., 2014), speech recognition (Amodei et al., 2015), and reinforcement learning (Hausknecht and Stone, 2015). Most RNNs, including popular variants such as long short-term memories (LSTMs), introduced by Hochreiter and Schmidhuber (1997), and gated recurrent units (GRUs), introduced by Cho et al. (2014), contain a non-linear dependency between sequential inputs. These non-linear dependencies create a very flexible class of models but limit the feasibility of training RNNs on long sequences as each sequence element must be processed sequentially. Modelling sequences of thousands to millions of elements is important to domains such as robotics, remote sensing, control systems, speech recognition, medicine, and finance.

The RNN serial evaluation inefficiency problem is usually mitigated by parallelizing the forward and backward pass over a minibatch of inputs. Without minibatches, RNN evaluation is a sequence of matrix-vector multiplications. Minibatches transform RNN computation into a sequence of more efficient matrix-matrix multiplications, but this speed-up brings several disadvantages. RNN model size is often limited by GPU memory size, and running a forward and backward pass on a minibatch requires memory linear in the minibatch size. Grouping data into minibatches increases the latency of each pass and reduces the rate of optimization steps. Finally, training with larger minibatches damages generalization ability (Keskar et al., 2017). Given these effects, it is desirable to obtain high training throughput with small minibatches. Persistent RNNs (Diamos et al., 2016) use a novel implementation that can achieve high GPU utilization with very small minibatch sizes when the recurrent state is larger than 500 elements, but even persistent RNNs become limited by the serial evaluation inefficiency at smaller hidden sizes.

Numerous prior works have shown strong performance from neural sequential models with only linear dependence on earlier sequence elements. Balduzzi and Ghifary (2016) investigated RNNs with only elementwise linear recurrence relations $h_t = \alpha_t \odot h_{t-1} + (1 - \alpha_t) \odot x_t$ and developed linear

---

*Currently at the Future of Humanity Institute, University of Oxford, Oxford, UK

variants of LSTM and GRU that perform similarly to standard non-linear RNNs on text generation tasks. Bradbury et al. (2017), Kalchbrenner et al. (2016), Gehring et al. (2017), and van den Oord et al. (2016) have successfully applied networks of convolutions over sequences for tasks such as machine translation, language modelling, and audio generation. These works have observed up to an order of magnitude increase in training throughput compared to RNN alternatives. Convolutional sequence models typically rely on either an attention mechanism or a (possibly linear) recurrent layer to integrate information at scales larger than the filter width. Introduction of a recurrent layer prevents full parallelization over the sequence length while attention mechanisms are expensive to apply on long sequences in online inference use cases.

A linear recurrence is a specific instance of a general form of computation known as a scan. Scans and reductions are computations involving repeated application of a binary operator $\oplus$ over an array of data. Computing the sum or maximum of an array is an example of a reduction, while a cumulative sum is a common example of a scan operation. Throughout this work, the scan of $\oplus$ with initial value $b$ is defined as

$$\text{SCAN}(\oplus, [a_1, a_2, ..., a_n], b) = [(a_1 \oplus b), (a_2 \oplus a_1 \oplus b), ..., (a_n \oplus a_{n-1}... \oplus a_1 \oplus b)].$$

The reduction of $\oplus$ over array $A$ and initial value $b$ is denoted $\text{REDUCE}(\oplus, A, b)$ and is the final element of $\text{SCAN}(\oplus, A, b)$. Despite their dependent computation graph, algorithms exist to parallelize scans and reductions when $\oplus$ is associative (Ladner and Fischer, 1980).

Blelloch (1990) shows that first order recurrences of the form $h_t = (\Lambda_t \otimes h_{t-1}) \oplus x_t$ can be parallelized with the parallel scan algorithm if three conditions are met:

1. $\oplus$ is associative: $(a \oplus b) \oplus c = a \oplus (b \oplus c)$
2. $\otimes$ is semiassociative: there exists a binary associative operator $\odot$ such that $a \otimes (b \otimes c) = (a \odot b) \otimes c$
3. $\otimes$ distributes over $\oplus$: $a \otimes (b \oplus c) = (a \otimes b) \oplus (a \otimes c)$

Considering the familiar operations in linear algebra, we see that the associative operation of vector addition ($x \oplus y = x + y$), the semiassociative operation of matrix-vector multiplication ($A \otimes x = Ax$) and the associative operation of matrix-matrix multiplication ($A \odot B = AB$) satisfy Blelloch's three conditions, allowing $h_t = \Lambda_t h_{t-1} + x_t$ to be evaluated in parallel over time steps $t$ for vectors $x_t$ and square matrices $\Lambda_t$.

We investigate this idea further and deliver the following contributions:

- We classify RNNs which satisfy the conditions above, and show that many RNNs used in practice such as the Quasi-RNNs (QRNNs) introduced by Bradbury et al. (2017) are contained in this class.
- We provide an implementation of the parallel linear recurrence algorithm as a CUDA kernel, and show that it speeds up training of QRNN and Lei and Zhang (2017)'s Simple Recurrent Unit (SRU) architectures by factors of up to 9x.
- We describe how several recent linear RNNs can be described as linear surrogates for non-linear architectures. We introduce a linear surrogate for the LSTM and show that we are able to train it with a speedup of 5-10x compared to the CuDNN LSTM when we use the parallel linear recurrence algorithm.

## 2 PARALLEL LINEAR RECURRENCE

As the method is essential to this work, Algorithm 1 presents the parallel linear recurrence algorithm for the interested reader.

### 2.1 THEORETICAL PERFORMANCE

The cost of a serial scan over a sequence of length $T$ is $C_{\text{sscan}} \in \mathcal{O}((C_\otimes + C_\oplus)T)$, compared to the parallel scan cost $C_{\text{pscan}} \in \mathcal{O}(2(C_\odot + C_\otimes + C_\oplus)(T/p + \lg p))$ on $p$ processors (Blelloch, 1990). If $h_t$ is a vector of dimension $n$ then $C_\odot \in \mathcal{O}(n^3), C_\otimes \in \mathcal{O}(n^2), C_\oplus \in \mathcal{O}(n)$ giving

---

**Algorithm 1** Parallel linear recurrence on $p$ processors

---
1: Let $y = [(\Lambda_1, x_1), (\Lambda_2, x_2), ..., (\Lambda_T, x_T)]$
2: Let binary operator $\bullet$ act as $(\Lambda, x) \bullet h = \Lambda h + x$
3: Let $S_0 = 1, S_i < E_i, E_i + 1 = S_{i+1}, E_{p-1} = T$ for $i$ in $0, p-1$
4:
5: **parfor** $i \leftarrow 0, p-1$ **do**
6:     $P_i = \text{REDUCE}(\odot, \Lambda_{S_i:E_i}, I)$
7:     $R_i = \text{REDUCE}(\bullet, y_{S_i:E_i}, 0)$
8: **end parfor**
9:
10: Let $z = [(P_0, R_0), (P_1, R_1), ..., (P_p, R_p)]$.
11: $C = \text{SCAN}(\bullet, z, h_0)$                      $\triangleright$ compute $C_i = P_i C_{i-1} + R_i$ with $C_{-1} = h_0$
12:
13: **parfor** $i \leftarrow 0, p-1$ **do**
14:     $h_{S_i:E_i} = \text{SCAN}(\bullet, y_{S_i:E_i}, C_{i-1})$
15: **end parfor**
16: **return** $h$

---

$C_{\text{pscan}} \in \mathcal{O}(2(n^3 + n^2 + n)(T/p + \lg p))$ and $C_{\text{sscan}} \in \mathcal{O}((n^2 + n)T)$. The $\mathcal{O}(n^3)$ cost of the matrix multiplication in the parallel algorithm can counter-act any parallel speedups for sufficiently large hidden states and lead to a slower algorithm overall.

To avoid this problem, we will only consider diagonal matrices $\Lambda_t$, in which case both matrix-matrix and matrix-vector multiplication have cost proportional to $n$ and $C_{\text{pscan}} \in \mathcal{O}(6n(T/p + \lg p))$ and $C_{\text{sscan}} \in \mathcal{O}(2nT)$. This gives a parallel speedup factor of $pT/3(T + \lg p)$. Assuming $p \ll T$, then $C_{\text{pscan}} \leq C_{\text{sscan}}$ when $p \geq 3$.

As we are only considering diagonal matrices, we write the linear recurrence as $h_t = \lambda_t \odot h_{t-1} + x_t$ where $\odot$ indicates elementwise multiplication.

Limiting $\Lambda_t$ to be diagonal may seem like a severe constraint but there are several reasons to do so beyond the favorable parallelization performance. Relatively few neural network models use separate recurrent matrices for each sequence element and using these separate matrices would require potentially prohibitive $n^2 T$ memory. Applying the same matrix $\Lambda$ to each sequence element is also unappealing considering that a matrix multiplication can be thought of as a rotation and a scaling. The same rotation at every element seems unlikely to be useful, and the scaling is exactly what's captured in diagonal vectors $\lambda_t$. Recurrent coefficient vectors $\lambda_t$ provide enough flexibility to implement schemes such as exponential moving averages or a gating mechanism.

## 2.2 BACKPROPAGATION

$$\nabla_{h_T} L = \frac{\partial L}{\partial h_T}$$

$$\nabla_{h_t} L = \frac{\partial h_{t+1}}{\partial h_t} \odot \nabla_{h_{t+1}} L + \frac{\partial L}{\partial h_t}$$

$$= \lambda_{t+1} \odot \nabla_{h_{t+1}} L + \frac{\partial L}{\partial h_t}$$

$$\nabla_{\lambda_t} L = \frac{\partial h_t}{\partial \lambda_t} \odot \nabla_{h_t} L = h_{t-1} \odot \nabla_{h_t} L$$

$$\nabla_{x_t} L = \nabla_{h_t} L$$

$$\nabla_{h_0} L = \frac{\partial h_1}{\partial h_0} \odot \nabla_{h_1} L = \lambda_1 \odot \nabla_{h_1} L$$

The backpropagation equations center around a linear recurrence over $\frac{\partial L}{\partial h_t}$ in the reverse order of the original sequence. This allows for parallelizing both the forwards and backwards pass of a linear RNN over the sequence length.

## 2.3 IMPLEMENTATION

GPUs commonly used for deep learning in 2017 consist of between 640 and 3200 parallel processors known as warps. Each warp operates on 32 single precision floating point numbers in parallel.

This work implemented parallel linear recurrence as a CUDA kernel with bindings into the TensorFlow (Abadi et al., 2016) framework. Each warp acts as a processor, which means the algorithmic $p$ is up to 3200 and the theoretical parallelization speedup factor is up to several hundred. The 32 lanes of each warp work on different elements of the recurrence vector in parallel. These implementation details mean that peak performance is only obtained on sequences of at least several thousand steps on at least a 32 element vector.

The parallel linear recurrence CUDA kernel and TensorFlow bindings are available at `https://github.com/eamartin/parallelizing_linear_rnns`.

## 3 MODELS

Parallel linear recurrence can be used to construct a wide variety of differentiable modules that can be evaluated in parallel. Common applications of linear recurrence include gating schemes and exponential moving averages. Although linear recurrence values can depend only linearly on previous elements, the stacking of linear recurrent layers separated by non-linearities allows for a non-linear dependence on the past. In this sense the non-linear depth of a linear recurrent network is the number of layers and not the sequence length.

### 3.1 GATED IMPULSE LINEAR RECURRENT LAYER

A gated impulse linear recurrent (GILR) layer transforms its $m$ dimensional inputs $x_t$ into a sequence of $n$ dimensional hidden states $h_t$:

$$g_t = \sigma(Ux_t + b_g)$$
$$i_t = \tau(Vx_t + b_z)$$
$$h_t = g_t \odot h_{t-1} + (1 - g_t) \odot i_t$$

A GILR layer applies the same non-linear transform to each sequence element and then accumulates the sequence elements with a non-linear gating mechanism. Gate $g_t$ uses the sigmoid activation function to give values in [0,1] for reasonable gating semantics, while impulse $i_t$ can use any activation function $\tau$. Stacking GILR layers allows for rich non-linear dependence on previous events while still taking advantage of fast parallel sequence evaluation.

#### 3.1.1 IMPACT ON EFFECTIVE "BATCH SIZE"

Consider evaluating an RNN with recurrence $h_t = \sigma(Uh_{t-1} + Vx_t + b)$ from $m$ inputs to $n$ hidden units on a sequence of length $T$ with minibatch size $b$ using a serial evaluation strategy. At each of $T$ iterations, the naive approach performs two $(b, m) \times (m, n)$ matrix multiplications. Larger matrix multiplications achieve higher throughput due to less IO overhead, so the better approach computes $Vx_t$ for all $t$ ahead of time in a single $(bT, m) \times (m, n)$ matrix multiply. The non-linear recurrence forces even the better approach to perform $T$ potentially small $(b, m) \times (m, n)$ matrix multiplications in serial. This makes serial RNN performance heavily dependent on minibatch size.

Now consider the GILR, noting that it has the same two matrix-vector multiplications per iteration as the above RNN. The intermediate variables $g$ and $i$ can be evaluated for all $t$ with a single $(bT, m) \times (m, n)$ matrix multiplication each. Given $g$ and $i$, $h$ can be computed using a parallel linear recurrence over $T$ vectors each of $bn$ elements. Rather than $T$ small operations, the GILR can be evaluated over all sequence elements with two large matrix multiplications and a parallel linear recurrence. GILR performance is much less dependent on batch size as the matrix multiplication kernel sees an "effective batch size" of $bT$ and $T$ is typically large.

### 3.2 LINEAR SURROGATE RNNS

RNNs learn a transition function $s_t = f(s_{t-1}, x_t)$ which combines previous state $s_{t-1}$ with input $x_t$ to compute current state $s_t$. Non-linear $f$ prevents application of the parallel linear recurrence

algorithm and forces slow serial evaluation. To work around this inefficiency, note that $s_t$ serves dual purposes. In $s_t = f(s_{t-1}, x_t)$, $s_{t-1}$ serves as an input to $f$ summarizing the previous inputs while $s_t$ serves as the output of $f$ to be passed to other layers of the network. We can decouple these uses and introduce independent variables for each purpose: $s_t$ is passed onto other layers of the network and we introduce the linear surrogate $\tilde{s}_t$ which is passed onto the next state, with $s_t = f(\tilde{s}_{t-1}, x_t)$. We are still able to choose a non-linear $f$, our only limitation being that $\tilde{s}_t$ must be linearly computable. We refer to this class of model as a linear surrogate RNN (LS-RNN). QRNNs (Bradbury et al., 2017) are LS-RNNs using $\tilde{h}_{t-1} = W_k x_{t-k} + ... W_1 x_{t-1}$ and strongly typed RNNs (Balduzzi and Ghifary, 2016) are LS-RNNs with $\tilde{h}_t = x_{t-1}$. Although not a rule, LS-RNNs can often be parallelized over sequence length with either convolution or linear recurrence.

Consider an LSTM:

$$f_t, i_t, o_t = \sigma(U_{f,i,o} h_{t-1} + V_{f,i,o} x_t + b_{f,i,o})$$
$$z_t = \tau(U_z h_{t-1} + V_z x_t + b_z)$$
$$c_t = f_t \odot c_{t-1} + i_t \odot z_t$$
$$h_t = o_t \odot c_t$$

An LSTM has state $s_t = (h_t, c_t)$. Since $c_t$ depends only linearly on $c_{t-1}$, no surrogate is needed for $c_t$. $h_t$ has a non-linear dependence on $h_{t-1}$, so $h_t$ needs a linear surrogate. Introducing a GILR layer as the surrogate, we obtain the GILR-LSTM:

$$g_t = \sigma(V_g x_t + b_g)$$
$$j_t = \tau(V_j x_t + b_j)$$
$$\tilde{h}_t = g_t \odot \tilde{h}_{t-1} + (1 - g_t) \odot j_t$$
$$f_t, i_t, o_t = \sigma(U_{f,i,o} \tilde{h}_{t-1} + V_{f,i,o} x_t + b_{f,i,o})$$
$$z_t = \tau(U_z \tilde{h}_{t-1} + V_z x_t + b_z)$$
$$c_t = f_t \odot c_{t-1} + i_t \odot z_t$$
$$h_t = o_t \odot c_t$$

For $m$ inputs and hidden size $n$, a GILR-LSTM contains $2n(n + m)$ more parameters than the equivalently sized LSTM to handle the mapping from $x$ to $\tilde{h}$. More generally, a LS-RNN contains all of the same parameters as the underlying RNN as well as some additional parameters to compute the linear surrogate.

## 4 EXPERIMENTS

We perform several experiments. First we find that our parallel linear recurrence kernel is able to achieve up to 40x higher throughput than a serial implementation when applied to long sequences. Secondly, we confirm that this kernel speedup translates to up to a 9x speedup to LS-RNNs such as QRNNs.

In order to illustrate that the linearization does not necessarily come at the cost of expressibility, we show that the GILR-LSTM architecture computed with the parallel linear recurrence algorithm is able to train significantly faster than an optimized LSTM implementation on a pathological long-term dependency problem from the original LSTM paper (Hochreiter and Schmidhuber, 1997).

### 4.1 THROUGHPUT BENCHMARKS

#### 4.1.1 KERNEL PERFORMANCE

We first illustrate the throughput advantage of the parallel scan algorithm for evaluating the linear recurrence. For a minibatch comprised of $b$ sequences of length $T$, we define the number of events as $bT$ and the throughput as the number of events processed per second. We implement two CUDA

Table 1: Parallel kernel speedup on $m$ features (minibatch size $= 1$)

| Sequence Length | $m = 4$ | $m = 32$ | $m = 128$ |
|---|---|---|---|
| 16 | 0.06 | 0.06 | 0.05 |
| 256 | 0.22 | 0.22 | 0.86 |
| 4,096 | 1.02 | 2.94 | 3.36 |
| 65,536 | 38.5 | 41.8 | 17.5 |

Table 2: Parallel kernel speedup for a variety of LS-RNNs, implemented as two stacked RNN layers with 256 hidden units. We keep the GPU memory usage constant by fixing $bT = 65,536$ for minibatch size $b$ and sequence length $T$

| Sequence Length | SRU | QRNN (filter size 2) | QRNN (filter size 10) | GILR-LSTM |
|---|---|---|---|---|
| 16 | 0.28 | 0.38 | 0.78 | 0.61 |
| 256 | 0.84 | 0.86 | 0.99 | 0.91 |
| 4,096 | 1.38 | 1.18 | 1.05 | 0.98 |
| 65,536 | 9.21 | 6.68 | 2.05 | 1.41 |

kernels, one which evaluates the parallel linear recurrence described in algorithm 2, and one which evaluates the same linear recurrence on GPU in serial over sequence length and in parallel over features and minibatch. The performance of each kernel depends on two factors: the sequence length and the product of number of features and minibatch size. The performance measurements for this experiment are made directly at the kernel level, avoiding any overhead from TensorFlow. We find that the parallel kernel has a distinct advantage at long sequence lengths with a speedup factor of up to 40x, as shown in table 1. The parallel kernel does not perform well at short sequence lengths due to the overhead of multiple passes over data and communication between processors.

### 4.1.2 ACCELERATING EXISTING RNN ARCHITECTURES

Several recently introduced LS-RNNs can be accelerated with the parallel linear recurrence algorithm. We implemented SRUs, QRNNs (with filter width 2 and 10), and GILR-LSTMs that can be computed with either the standard serial linear recurrence algorithm or parallel linear recurrence. Both methods compute an identical recurrence, so switching from a serial to parallel implementation does not cause any numerical changes and takes only a single line of code changes. Notably, both SRUs and QRNNs claim an order of magnitude speedup compared to CuDNN LSTM when implemented with serial linear recurrence. Any further speedup from parallel linear recurrence applies on top of the existing speedup. We timed train throughput (forwards and backwards propagation), but the linear time of each pass also makes the results applicable to forwards (inference) performance. However, parallel linear recurrence can only accelerate inference in scenarios where the entire input sequence is known at the start of the inference phase. We controlled for GPU memory usage within these experiments by fixing $bT = 65,536$ for minibatch size $b$ and sequence length $T$, and chose a popular architecture consisting of two stacked RNN layers with 256 hidden units and an input size of 4.

Table 2 shows that the throughput advantage from using parallel linear recurrence compared to serial linear recurrence reaches up to 9x. Simpler architectures (for which the linear recurrence is a higher proportion of the total computational load) are more affected by the switch to the parallel kernel. This is particularly clear in the case of the QRNN, where including wider convolutional filters results in more time spent outside of the linear recurrence and therefore reduces the speedup from linear recurrence parallelization.

### 4.2 SYNTHETIC EXPERIMENT

One of the key strengths of the LSTM is that it is capable of dealing with long-term dependencies. In order to demonstrate that the GILR-LSTM is also able to handle long-term dependencies we tackle a canonical example of inference over many time steps from Hochreiter and Schmidhuber (1997). We show that in fact the GILR-LSTM is able to outperform the CuDNN LSTM and extend to sequence

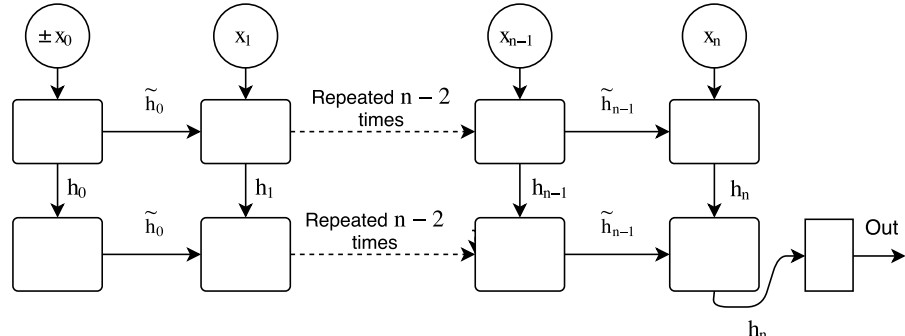

Figure 1: The structure of the synthetic example and the GILR-LSTM architecture we used to tackle it. We feed in one-hot unit vectors $x$ which are chosen uniformly at random (with replacement). The class is determined by the very first vector $x_0$, which has a fixed direction. The sign of $x_0$ determines the class. In the diagram, each rounded block indicates a cell of the RNN, whilst the square indicates a linear unit.

Table 3: Performance of the GILR-LSTM compared to the CuDNN LSTM on problem 2b from Hochreiter and Schmidhuber (1997).

| Sequence Length | 1,024 | | 8,192 | | 1,048,576 | |
|---|---|---|---|---|---|---|
| | CuDNN | GILR | CuDNN | GILR | CuDNN | GILR |
| Iterations (1000s) | $1.0 \pm 0.4$ | $0.55 \pm 0.04$ | $0.44 \pm 0.05$ | $0.56 \pm 0.16$ | - | $14 \pm 3$ |
| Wall time (hours) | $0.28 \pm 0.08$ | $0.031 \pm 0.002$ | $0.58 \pm 0.06$ | $0.10 \pm 0.03$ | - | $9.7 \pm 1.7$ |

lengths orders of magnitude longer than dealt with previously. The input consists of sequences of length $n$ where for $n > 0$ each element is a randomly chosen one-hot vector $x$ in $p$-dimensional space. The first vector in each sequence, $x_0$, is always either $(1, 0, \ldots, 0)$ or $(-1, 0, \ldots, 0)$. The sequential model must read in an entire sequence and then output the sign of the first sequence element. This sequence classification problem requires remembering the first element over the length of the sequence, and early RNNs struggled with this for $p$ as small as a few dozen. In the original formulation of the problem (dealing in the regime with around one hundred timesteps), the dimensionality of the input $p$ is set equal to $n$. Since this would make the size of the input data grow impractically large as $\mathcal{O}(n^2)$ for long sequences, we fix $p = 128$ as we vary $n$. We generated sequences for $n$ equal to 1,024, 8,192, and 1,048,576. For each of these we compared a two layer GILR-LSTM with 512 hidden units to a two layer LSTM with 512 hidden units[1] per layer implemented by CuDNN.

We ran all experiments on a NVIDIA K80 GPU, with five runs per configuration allowing us to find the average and standard deviation of the time and number of iterations to convergence. We continually generated random sequences to serve as input data. A brief search over learning rate and batch size was carried out to find the parameters which allow the network to converge most rapidly for all runs. The criterion for convergence was five consecutive minibatches giving 100% accuracy. The learning curves in figure 2 give support to this being a reasonable convergence criteria. For the longest sequence length, we did not observe the CuDNN LSTM converging, even after several days' training.

The results as show in table 4.2 illustrate that the GILR-LSTM is able to converge between 6 and 10 times faster than the CuDNN LSTM. This is somewhat surprising given the LSTM was specifically constructed for problems of this sort, and the CuDNN LSTM implementation is highly optimized (to the extent that the monolithic interface it exposes is difficult to modify or extend). The GILR-LSTM is implemented entirely in standard TensorFlow with the exception of using the new linear recurrence op instead of a TensorFlow symbolic loop. Convergence of the GILR-LSTM models leads to the conclusion that the non-linearities present in LSTM are not necessary for solving this instance of

---

[1]For the longest sequence length, the number of hidden units was decreased to 64 for both architectures so that the net could fit in memory.

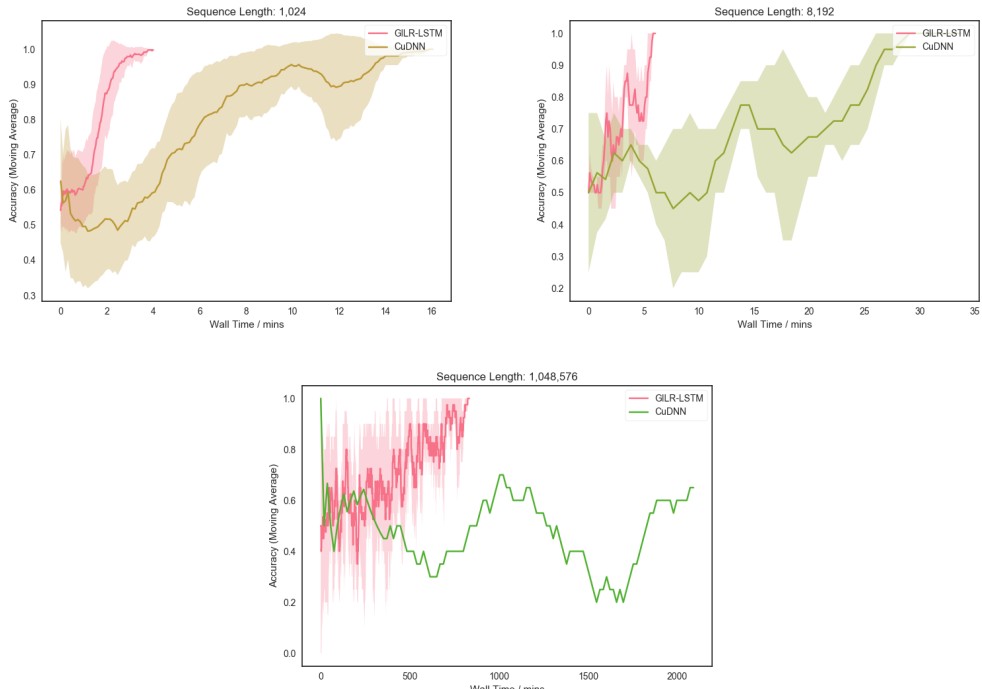

Figure 2: Learning curves for GILR-LSTM and CuDNN LSTM architectures for various sequence lengths. Each plot shows the moving mean and standard deviation of classification accuracy over five training runs, with the exception of a single run for CuDNN LSTM on 1 million sequence length.

the long-term dependency problem. The time to convergence further leads to the conclusion that inclusion of a non-linearity at every step incurs a significant training time slowdown. Furthermore, the GILR-LSTM is able to learn to carry dependencies over a one million element sequence. As far as we know, this one million step sequence experiment is the longest sequential learning problem to be handled by neural networks to date.

## 5 CONCLUSION

A significant portion of the success of deep learning can be attributed to access to massive amounts of computation. Most of this computation is accessed through two highly efficient and parallelizable building blocks: matrix multiplication and convolution. Recent research has demonstrated that linear RNNs can achieve similar prediction accuracy to non-linear RNNs on a wide variety of tasks in a fraction of the training time. We propose the framework of LS-RNNs as a way to tame the growing zoo of sequential neural nets. We identify linear recurrence as another parallelizable building block for current and future sequential models and we use it to obtain significant speedups on already fast models. With the power of parallel linear recurrence we are able to solve a sequential dependency problem multiple orders of magnitude larger than anything done prior. Future applications of parallel linear recurrence within neural nets could include parallel training of memory augmented models or providing a new sort of image filter on very high resolution images. We hope that parallel linear recurrence can be to large scale sequence modelling what fast convolution algorithms are to image recognition.

### ACKNOWLEDGMENTS

We would like to acknowledge Kevin Bowers, Alex Meiburg, JD Co-Reyes, Carson McNeil, Andy Palan, Sören Mindermann, and several others for fruitful conversations and guidance.

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
