# OpenReview forum: "Parallelizing Linear Recurrent Neural Nets Over Sequence Length"
_ICLR.cc/2018/Conference — Accept (Poster)_

### Official Review · AnonReviewer2 · 2017-11-26
**simple but effective method for RNN speed up**

**Rating:** 6
**Confidence:** 3

**Review:**

This paper focuses on accelerating RNN by applying the method from Blelloch (1990). The application is straightforward and thus technical novelty of this paper is limited. But the results are impressive.

One concern is the proposed technique is only applied for few types of RNNs which may limit its applications in practice. Could the authors comment on this potential limitation?

---

> ### Author Response · Authors · 2017-12-08
> **Significant quantity of research indicates non-linear recurrence not necessary for sequence problems**
>
> We contest the limited technical novelty of this work. It is true that parallel scan is "a key primitive in many parallel algorithms"[1] and has been heavily studied and optimized. Parallel linear recurrence is a lesser known application of the widely popular parallel scan algorithm. Neural nets are hugely dependent on high performance parallel computational primitives such as matrix multiplication and convolution. We believe the first application of this classic parallel algorithm to a field dependent on fast parallel algorithms is a novel idea; otherwise someone else would have published this paper in the previous 30+ years that both parallel linear recurrence and RNNs have existed.
>
> Beyond the new architectures introduced in the paper, we applied parallel linear recurrence (PLR) to SRU and QRNN and note that it could also be applied to strongly-typed RNNs. Further, we show that PLR can also accelerate (the currently uninvestigated) architectures involving on h_t = A_t h_{t-1} + x_t for square matrices A_t.
>
> The broader question is "how limiting is it that PLR cannot accelerate LSTMs, GRUs, vanilla RNNs, or other non-linear RNN models?". We do not think this will limit the applicability of PLR within RNNs. A significant amount of recent research (listed below in [2]) has matched or surpassed the performance of non-linear RNNs with models with only linear sequential dependency. Given this body of research, our belief has shifted from "RNNs depend on sequential non-linearity" to "there is no evidence that sequential non-linearity is necessary, and there is a fair amount of evidence it is not necessary". With this in mind, we believe PLR's incompatibility with non-linear RNNs is not a major practical limitation as we expect linear surrogate RNNs to continue growing in popularity due to their fast training times and good performance. We also think this work will accelerate the growing popularity of linear surrogate RNNs.
>
> [1]
> http://people.cs.vt.edu/yongcao/teaching/cs5234/spring2013/slides/Lecture10.pdf
>
> [2]
> Sequential models with linear dependendencies with experimental
> performance on par with non-linear RNNs. Most models listed trained in
> significantly less time than non-linear RNN.
>
> Strongly-typed RN https://arxiv.org/abs/1602.02218 (language
> modelling)
>
> ByteNet https://arxiv.org/abs/1610.10099 (state of the art (SotA)
> character level language model on Hutter Prize, SotA character to
> character machine on WMT)
>
> Quasi-RNN https://arxiv.org/abs/1611.01576 (sentiment classification,
> language modelling, machine translation)
>
> Convolutional Sequence to Sequence Learning
> https://arxiv.org/abs/1705.03122 (machine translation, outperforms
> LSTM)
>
> Attention Is All You Need https://arxiv.org/abs/1706.03762 (SotA
> machine translation on WMT)
>
> WaveNet https://arxiv.org/abs/1609.03499 (high fidelity audio
> generation)
>
> Simple Recurrent Unit https://arxiv.org/abs/1709.02755 (matches or
> outperforms LSTM on sequence classification, question answering,
> language modelling, machine translation, speech recognition). PLR
> can significantly accelerate already fast SRU training.

---

### Official Review · AnonReviewer1 · 2017-11-27
**Authors propose a method to make recurrent learning over 1000s and more time steps possible.**

**Rating:** 7
**Confidence:** 2

**Review:**

# Summary and Assessment

The paper addresses an important issue–that of making learning of recurrent networks tractable for sequence lengths well beyond 1’000s of time steps. A key problem here is that processing such sequences with ordinary RNNs requires a reduce operation, where the output of the net at time step t depends on the outputs of *all* its predecessor.
The authors now make a crucial observation, namely that a certain class of RNNs allows evaluation in a non-linear fashion through a so-called SCAN operator. Here, if certain conditions are satisfied, the calculation of the output   can be parallelised massively.
In the following, the authors explore the landscape of RNNs satisfying the necessary conditions. The performance is investigated in terms of wall clock time. Further, experimental results of problems with previously untacked sequence lengths are reported.

The paper is certainly relevant, as it can pave the way towards the application of recurrent architectures to problems that have extremely long term dependencies.
To me, the execution seems sound. The experiments back up the claim.

## Minor
- I challenge the claim that thousands and millions of time steps are a common issue in “robotics, remote sensing, control systems, speech recognition, medicine and finance”, as claimed in the first paragraph of the introduction. IMHO, most problems in these domains get away with a few hundred time steps; nevertheless, I’d appreciate a few examples where this is a case to better justify the method.

---

> ### Author Response · Authors · 2017-12-08
> **Example tasks for which truncated backprop through time causes problems**
>
> We agree that you can often "get away with" backprop through time
> (BPTT) truncated at several hundred time steps for many sequential
> problems, even when the inherent sequence length of the data is very
> long.
>
> Some problems which can benefit from additional sequence length:
>
> * Medical waveforms are often sampled at greater than 1KHz. This means
>   relatively short recordings create very long sequences. These
>   sequences may be used for a sequence classification task which makes
>   it difficult to use truncated BPTT. Sequence classification on very
>   long sequences must either handle the entire sequence, classify
>   subsequences (suboptimal as label may only be determined by part of
>   the sequence), or down-sample the sequence data (suboptimal because
>   it loses information). The 2016 PhysioNet Challenge
>   (https://physionet.org/challenge/2016/) involved classifying EEGs
>   sampled at 2KHz for 5-120s for a total of 10K-240K events per
>   sequence. It would be difficult to apply neural nets to such a
>   problem without a technique to parallelize over timesteps.  An even
>   more extreme dataset is 90 minutes @ 30KHz (= 160 million steps) of
>   neural recordings of a mouse: http://data.cortexlab.net/dualPhase3/ .
>   In general, consider some task involving sensor data. Now consider the
>   same phenomena but measured at 10X the frequency. There is now
>   more information available, but this additional information is only accessible if
>   the researcher has tools that can deal with a 10X longer sequence with
>   10X longer dependencies.
>
> * Example future machine learning task: Generate a (text) review of a
>   2+ hour movie, including comments on dialogue and
>   cinematography. Even with significant downsampling of both frames
>   and audio, a 2 hour movie contains 7200 frames at 1 frame/sec and an
>   average of 9000 words
>   (http://kaylinwalker.com/long-winded-actors-and-movies-with-the-most-dialogue/).
>   We believe parallel sequential methods would be hugely useful for
>   such a task.
>
> * I am not an expert, but I believe reinforcement learning on long
>   episodes with sparse rewards could benefit from less episode truncation.

---

### Official Review · AnonReviewer3 · 2017-11-27
**Faster RNNs, with novel insights on need for nonlinear recurrence; novel and clear presentation**

**Rating:** 7
**Confidence:** 4

**Review:**

This paper abstracts two recently-proposed RNN variants into a family of RNNs called the Linear Surrogate RNNs which satisfy  Blelloch's criteria for parallelizable sequential computation. The authors then propose an efficient parallel algorithm for this class of RNNs, which produces speedups over the existing implements of Quasi-RNN, SRU, and LSTM. Apart from efficiency results, the paper also contributes a comparison of model convergence on a long-term dependency task due to (Hochreiter and Schmidhuber, 1997). A novel linearized version of the LSTM outperforms traditional LSTM on this long-term dependency task, and raises questions about whether RNNs and LSTMs truly need the nonlinear structure.

The paper is written very well, with explanation (as opposed to obfuscation) as the goal. Linear Surrogate RNNs is an important concept that is useful to understand RNN variants today, and potentially other future novel architectures.

The paper provides argument and experimental evidence against the rotation used typically in RNNs. While this is an interesting insight, and worthy of further discussion, such a claim needs backing up with more large-scale experiments on real datasets.

While the experiments on toy tasks is clearly useful, the paper could be significantly improved by adding experiments on real tasks such as language modelling.

---

> ### Author Response · Authors · 2017-12-08
> **Models which benefit from parallel linear recurrence have demonstrated strong experimental performance**
>
> We feel the very impressive performance of SRUs and QRNNs on a variety of large-scale tasks demonstrates the applicability and usefulness of our work. We could have replicated their results in language modelling and machine translation with faster training times, but we believe that showing large speedup factors for these models is sufficient evidence for the value of parallel linear recurrence.
>
> We argue more strongly that a non-linearity recurrence is unnecessary than we do that "rotation free" RNNs are just as powerful as RNNs with non-diagonal weight matrices. However, SRUs are "rotation free" linear recurrences with performance equal or superior to LSTM and other non-linear RNNs on 6 sequence classification datasets, the SQuAD question answering dataset, Penn Treebank language modelling, Switchboard-1 speech recognition, and WMT English->German translation.

---

### Decision · Program_Chairs · 2018-01-29
**ICLR 2018 Conference Acceptance Decision**

**Decision:**

Accept (Poster)

**Comment:**

Paper presents a way in which linear RNNs can be computed (fprop, bprop) using parallel scan. They show big improvements in speedups and show application on really long sequences. Reviews were generally favorable.